

# Agrochemical control of gene expression using evolved split RNA polymerase

Yuan Yuan[1] and Jin Miao[2]

[1] Department of Neurophysiology and Neuropharmacology, Institute of Special Environmental Medicine and Co-innovation Center of Neuroregeneration, Nantong University, Nantong, Jiangsu Province, China
[2] Duke Kunshan University, Kunshan, China

## ABSTRACT

Chemically-inducible gene expression systems are valuable tools for rational control of gene expression both for basic research and biotechnology. However, most chemical inducers are confined to certain groups of organisms. Therefore, dissecting interactions between different organisms could be challenging using existing chemically-inducible systems. We engineered a mandipropamid-induced gene expression system (Mandi-T7) based on evolved split T7 RNAP system. As a proof-of-principle, we induced GFP expression in *E. coli* cells grown inside plant tissue.

## INTRODUCTION

Chemically-inducible gene expression systems are powerful tools that allow conditional control over gene expression and facilitate study of toxic and essential genes. However, most chemically-inducible systems were often designed to control gene expression in a single group of organisms in the laboratory condition. Working in a complex context, like the symbiosis between plant and bacteria, requires new chemical inducers that are neutral to different groups of organisms. Agrochemicals are attractive candidates because of their specific mode of action, low toxicity, and drug-like physical properties (*Delaney et al., 2006*). Mandipropamid is the active ingredient of oomyceticide REVUS® and has been repurposed to control drought tolerance of plants through orthogonal control strategy (*Park et al., 2015*). Mandipropamid and PYR1^MANDI protein constitute an engineered ligand-receptor pair, orthogonal to abscisic acid (ABA)-PYR1 receptor pair (*Park et al., 2015*). The mandipropamid-PYR1^MANDI pair binds protein phosphatase HAB1 or ABI1 as the ABA-PYR1 pair (*Park et al., 2015*, *Ziegler et al., 2022* and Fig. S1). RNA polymerases which are modulated by protein-protein interaction would be desirable for coupling with mandipropamid-sensing module to activate gene transcription (*Baumschlager, Aoki & Khammash, 2017*; *Han, Chen & Liu, 2017*; *Chee et al., 2022*). Split T7 RNA polymerase (RNAP) has been evolved to be a proximity-dependent biosensor platform, which could transduce protein-protein interaction of light or small molecule sensing modules into activation of gene expression under the control of T7 promoter (*Pu, Zinkus-Boltz & Dickinson, 2017*). Self-assembly of the evolved T7 RNAP system (eRNAP) has been largely

Corresponding author
Jin Miao,
jin.miao@dukekunshan.edu.cn

eliminated through series of directed evolution (*Pu, Zinkus-Boltz & Dickinson, 2017*; *Pu, Kentala & Dickinson, 2018*). This eRNAP platform enables detecting protein interaction (*Dewey & Dickinson, 2020*; *Lin et al., 2021*) and manipulating strength of interaction (*Zinkus-Boltz, DeValk & Dickinson, 2019*; *Dewey et al., 2021*). ABA inducible CRISPR/Cas9 gene editing in mammalian cells has been demonstrated using this evolved split T7 RNAP system (*Pu, Kentala & Dickinson, 2018*). Engineering mandipropamid induced gene expression is feasible by modifying abscisic acid sensing module. Here, we report the repurposing of mandipropamid to activate gene expression using the evolved split T7 RNA polymerase (Mandi-T7) and demonstrate the ability to regulate gene expression of bacteria grown inside plant tissue.

## MATERIALS AND METHODS

### Plasmid construction

All genetic parts were synthesized and cloned into vectors by Genscript Biotechnology Inc. (Nanjing, China). Plasmid pET23a(+) was used to clone the mandipropamid-T7 RNAP cassettes. And the backbone of compatible plasmid pCDFduet-1 was used to clone the T7pro- reporter cassette. Sequence of all genetic parts were described in Table S1 and Table S2.

### Mandipropamid responsive assay for evolved split T7 RNA polymerase

Both driver and reporter plasmids were co-transformed into *E. coli* strain Top10 (TIANGEN Biotech., Beijing, China). Single colonies were inoculated 2XYT medium with antibiotics at 37 °C. Overnight culture was transferred to fresh medium with antibiotics at 1:400 ratio. At the same time, mandipropamid (sc-235565, Santa Cruz, CA, USA) or DMSO (solvent control) was added. The culture was incubated for 6 h, and 100 μL of each sample was added to a 96 well plate and mixed with 190 μL 0.1% Tween-80. Both florescence (GFP, Ex: 488, Em: 510; mcherry, Ex: 587, Em: 630) and $OD_{600}$ was measured by the Perkin Elmer Enspire$^{TM}$ 2,300 Multilabel Reader. After the values for medium were subtracted, florescence over $OD_{600}$ was calculated and compared to that of the DMSO control samples.

### Mandipropamid responsive assay for split GFP

Plasmids were transformed into *E. coli* strain BL21(DE3) (TRANSGEN Biotech., Beijing, China). Single colonies were inoculated 2XYT medium with antibiotics at 37 °C. Overnight culture was transferred to fresh medium at 1:100 ratio. When $OD_{600}$ reached between 0.4 to 0.6, IPTG (1 mm final) was added to induce expression of GFP1-9. Mandipropamid or DMSO (solvent control) was also added. The culture was incubated for 3 h, and 100 μL of each sample was transferred to a 96 well plate. Both GFP florescence (Ex: 488, Em: 510) and $OD_{600}$ was measured by the Perkin Elmer Enspire$^{TM}$ 2,300 Multilabel Reader. After the values for medium were subtracted, florescence over $OD_{600}$ was calculated and compared to that of the DMSO control sample.

## Time course fluorescence measurement

*E. coli* Top10 strain containing Mandi-T7 and reporter plasmids was generated as mentioned above. Single colonies were inoculated 2XYT medium with antibiotics at 37 °C. Overnight culture was transferred to fresh medium with antibiotics at 1:200 ratio. When $OD_{600}$ reached roughly 0.2, aliquots of 500 μL culture was distributed into individual wells of a deep-well plate or into the hollow septate stems of water spinach (*Ipomoea aquatica*). Mandipropamid (sc-235565, Santa Cruz, CA, USA) or DMSO (solvent control) was then added. The plate was incubated at 1,000 rpm for 6 h. Samples were harvested after 15 min, 30 min, 1 h, 2 h, 3 h, 4 h, and 6 h. A total of 100 μL of each sample was added to a 96 well plate. Both florescence (GFP: Ex: 488, Em: 510; mcherry: Ex: 587, Em: 630) and $OD_{600}$ was measured by Perkin Elmer Enspire™ 2,300 Multilabel Reader. After the values for medium were subtracted, florescence over $OD_{600}$ was calculated.

## Western blot analysis

*E.coli* culture was harvested at series of time points from deep-well plates using centrifuge. Samples were boiled in 1X loading buffer for 5 min. Relative protein concentration was determined by BCA assay. Total protein was resolved using SDS-PAGE. The protein sample was transferred to PVDF membranes (Millipore, Burlington, MA, USA) using Mini Trans-Blot (Bio-Rad, Hercules, CA, USA). Anti-GFP (MF090; 1:2,000 dilution) and anti-GAPDH (MF091; 1:1,000 dilution) antibodies from Mei5bio were used.

## Microscopic analysis of reporter expression in plant leaves

The *E. coli* Top10 strain containing Mandi-T7 and sfGFP reporter was generated as mentioned above. Overnight culture was transferred to fresh medium at 1:100 ration. When $OD_{600}$ reached between 0.4 to 0.6, *E. coli* cell was washed and resuspended with Murashige and Skoog medium (MS). Infiltration of *E. coli* cells into the abaxial side of *Arabidopsis* leaves was performed using a needless 1 mL syringe. Either mandipropamid (400 μm mandipropamid and 0.1% Silwet L-77) or mock (0.2% DMSO and 0.1% Silwet L-77) was applied on the abaxial surface of infiltrated leaves. After 3 h GFP fluorescence was imaged by Confocal microscope (Leica SP8). Representative images from three different positions were shown. A Z-stack of the same thickness was merged using the 'MAX' method (Leica LAS X). The fluorescence intensity and cell number were analyzed using ImageJ (version 1.53i).

## RESULTS

To simplify circuit design and troubleshooting, we deployed the Mandi-T7 driver module and reporter module on two compatible plasmids. For the reporter module, expression of superfolder green fluorescent protein (sfGFP) was driven by the T7 promoter. For the driver module, we followed the reported architecture of split T7 RNAP (*Pu, Kentala & Dickinson, 2018*). Briefly, we fused PYR1[MANDI] to the RNAP C-terminal half (T7 RNAPc) and the evolved RNAP N-terminal half (eRNAPn) to the N-terminal truncated ABI1 (Fig. 1A). We used relatively long flexible linker (12 aa) to fuse PYR1[MANDI] and T7 RNAPc

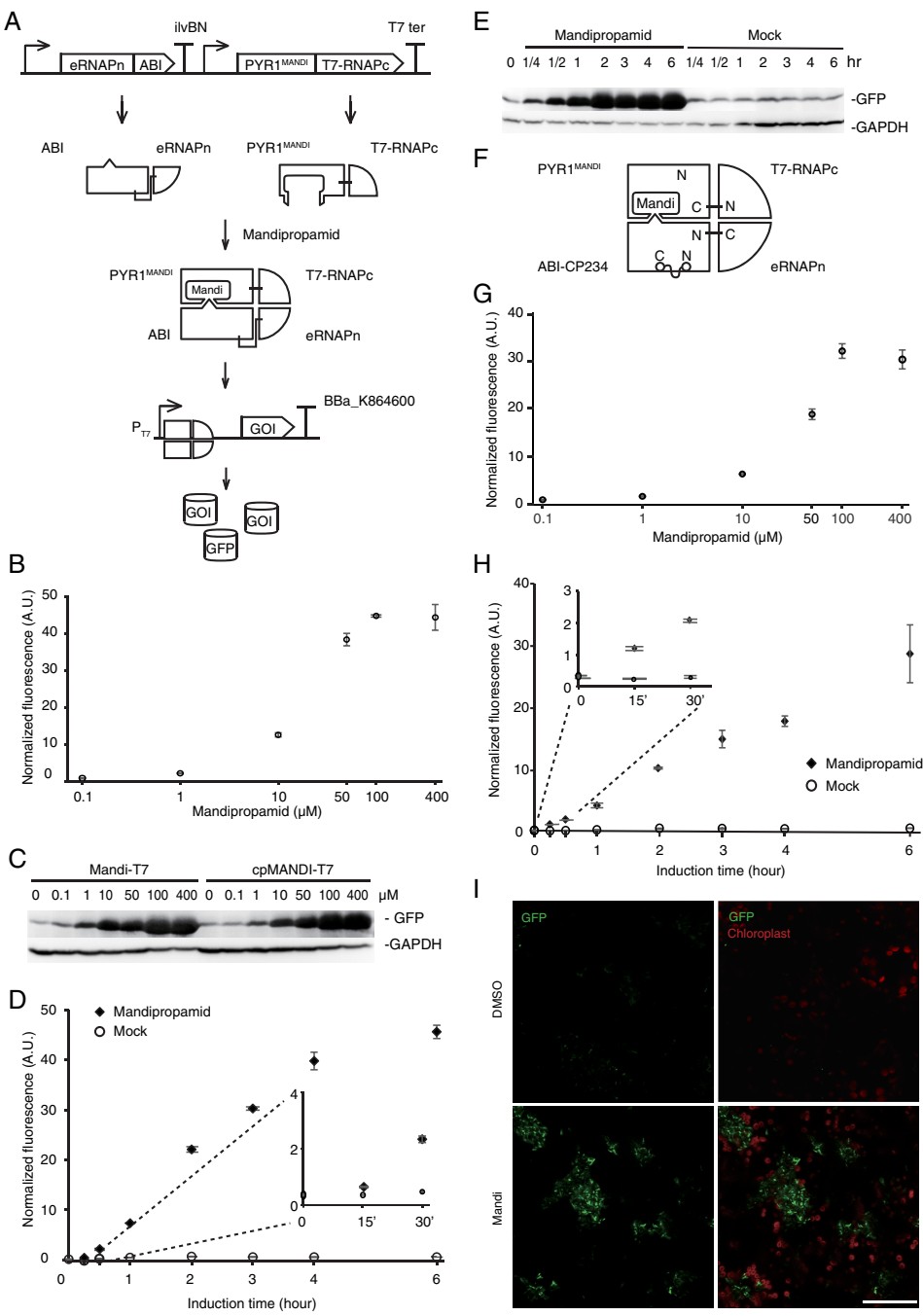

**Figure 1 Engineering the Mandi-T7 system and its application.** (A) Schematic of Mandi-T7 system and GFP reporter under control of T7 promoter. Bent arrow: promoter; Box arrow: CDS; Large T-shape: terminator. (B) Dose-response analysis of Mandi-T7 in *E. coli* Top10. (C) Western blot analysis of reporter expression in (B and G). (D) Inducer response after 6 h of induction in liquid culture. (E) Western blot analysis of reporter expression in (D). (F) Schematic of CP234-Mandi-T7. The original termini of ABI were linked by a flexible linker. (G) Dose-response analysis of CP234-Mandi-T7 in *E. coli* Top10. (H) Time course after induction for 6 h in the hollow septate stems of water spinach. (I) Exogenous control of *E. coli* gene expression inside leaves. Bar: 50 μm. The values in the scatter plots (B, D, G, and H) are averages of three biological replicates and the error bars represent standard deviation.

for achieving high-level expression upon induction (*Pu, Zinkus-Boltz & Dickinson, 2017*). We used the catalytic inactive mutant (D143A) of ABI1 protein (ABI) to minimize unintended effects (*Liang, Ho & Crabtree, 2011*). To avoid using additional inducer molecules, we used constitutive promoters to drive the expression of Mandi-T7. We chose standard promoter of medium strength (J23105) to drive the expression of PYR1$^{MANDI}$-T7 RNAPc to avoid overwhelming the metabolism (*Tan & Ng, 2020*). To evaluate the performance of Mandi-T7 system, we measured the response of the Mandi-T7 to different concentration of mandipropamid. The results showed a dose-dependent response, which saturated at 100 μm mandipropamid (Fig. 1B, Fig. 1C, and Fig. S2). A fold-induction of 44X was achieved.

To understand the kinetic characteristics of Mandi-T7 system, we measured fluorescence intensity over the course of induction for 6 h. Fluorescence above the background level was detected at 15 min (Fig. 1D and Fig. 1E). The induction plateaued after 6 h (Fig. 1D and Fig. 1E). We also detected fluorescence signal without induction (Fig. 1D and Fig. 1E). We also tested mcherry as the reporter. The dose-response curve of mcherry showed 140X induction at 100 μm (Fig. S3A). The time course expression analysis showed induction plateaued after 4 h (Fig. S3B). We failed to detect toxic effect of mandipropamid on *E.coli* growth (Fig. S4).

In a separate attempt, we tried a circular permutated version of ABI protein, which provides alternative topology for complementation of the split T7 RNAP. We noticed that Ser 234 at the end of helix 2 is closer to the C terminus of PYR1$^{MANDI}$ than the original N-terminus (Fig. S5). We joined the original N and C termini of ABI by a flexible linker and generated the circular permutant CP234, for which Ser234 of ABI serves as the new N-terminus (Fig. 1F). To test the retention of the activity to bind PYR1$^{MANDI}$, we used the tripartite split-GFP based system because of its low background fluorescence and minimal effect on fusion protein (*Cabantous et al., 2013*). Of the eleven β-strands in the β-barrel of GFP structure, two strands at the C-terminus are individually used as fusion tags to interacting protein partners and allow reconstitution with the rest of the GFP molecule (Fig. S6; *Cabantous et al., 2013*). We observed concentration dependent increase in GFP florescence after mandipropamid induction (Fig. S6). This result indicated that ABI-CP234 retained the ability to interact with PYR1$^{MANDI}$ in the presence of mandipropamid. Next, we adapted ABI-CP234 to the evolved split T7 RNAP system (CP234-Mandi-T7), replacing ABI using ABI-CP234. The result also showed strong sfGFP expression upon mandipropamid induction like the Mandi-T7 system, with a fold induction of 33 (Fig. 1C, Fig. 1G and Fig. S2).

To provide a proof-of-principle that Mandi-T7 is useful in complex setting, we evaluated the ability of Mandi-T7 to induce bacterial gene expression inside the hollow septate stems of water spinach (*Ipomoea aquatica*). We followed reporter expression using fluorescence intensity and Western blot. The result showed induction after 15 min and induction level kept on increasing over 6-h period (Fig. 1H and Fig. S8). To directly observe the induced reporter expression inside plant tissue, we infiltrated rosette leaves of *Arabidopsis* with *E.coli* culture harboring Mandi-T7 and sfGFP reporter and applied mandipropamid on the abaxial leaf surface. After 3 h, we assayed GFP florescence under

confocal microscopy. The autofluorescence of chloroplast in the plant cells provided a high-contrast background for observing GFP fluorescence. The result showed that under low excitation intensity, more GFP-positive *E.coil* cells could be observed inside the induced leaves than inside the control leaves (Fig. 1I). The intensity of GFP fluorescence was also higher after induction (Fig. S9). Basal level expression of GFP in the control groups was observable at higher excitation intensity and used here to count the *E.coli* cell number (Fig. S9). This result suggested that Mandi-T7 has the potential to enable manipulation of gene expression of plant-associated microorganisms.

## DISCUSSION

Our work established an inducible gene expression system, using agrochemical mandipropamid. The Mandi-T7 system achieved a fold-induction of 44X, within the same order of magnitude as the 100X induction of the ABA-inducible RNAP sensor at 100 μm (*Pu, Kentala & Dickinson, 2018*). The difference between the two systems is that ABA sensor can be further induced to 468X fold change at 10 mm (*Pu, Kentala & Dickinson, 2018*). The rapid activation of reporter gene expression shown in kinetic analysis is consistent with the nature of T7 RNA polymerase based system.

We have also tested mcherry as the fluorescence protein reporter. The increased fold change for mcherry is consistent with low cellular autofluorescence at its emission region (*Telford et al., 2012*).

The reporter signal in uninduced conditions could be caused by constitutive expression of Mani-T7 module and low level self-assembly of split T7 RNAP (*Pu, Kentala & Dickinson, 2018*). Further engineering efforts like directed evolution are needed to lower the basal expression level and increase the dynamic range.

We also tested alternative topology (ABI-CP234) for generating ABI fusion protein and hope this result might help expand mandipropamid to other proximity-dependent systems. CP234-ABI might also be applicable to ABA FRET sensor.

## CONCLUSIONS

This study provides a new tool for agrochemical control gene expression. We showed that mandipropamid inducible gene expression can be readily built based on the evolved split T7 RNAP system. T7 RNAP is orthogonal and widely applicable to many prokaryotic and eukaryotic organisms. Mandi-T7 can be further integrated with other tools like CRISPRi technology. Low toxicity, easy absorption, and cost-effectiveness of mandipropamid will give Mandi-T7 leverage in other complex contexts like vector-borne bacterial plant pathogens. We hope the Mandi-T7 system will expand rational control over gene expression to diverse biological context.

## ACKNOWLEDGEMENTS

We would like to thank Dr. Jinyue Pu and Professor Bryan Dickinson (University of Chicago) for sharing plasmids and vector information. We also want to thank Professor Linfeng Huang (Duke Kunshan University) for advice and generous sharing equipment and reagents.

### Funding
This work was supported by the National Natural Science Foundation of China (Grant 31500690 to Jin Miao). There was no additional external funding received for this study. The funders had no role in study design, data collection and analysis, decision to publish, or preparation of the manuscript.

### Grant Disclosures
The following grant information was disclosed by the authors:
National Natural Science Foundation of China: 31500690.

### Competing Interests
Yuan Yuan and Jin Miao filed an application for a patent (202110440791.X) on April 23, 2021.

### Author Contributions
- Yuan Yuan performed the experiments, analyzed the data, prepared figures and/or tables, authored or reviewed drafts of the article, and approved the final draft.
- Jin Miao conceived and designed the experiments, performed the experiments, analyzed the data, prepared figures and/or tables, authored or reviewed drafts of the article, and approved the final draft.

### Patent Disclosures
The following patent dependencies were disclosed by the authors:
Yuan Yuan and Jin Miao filed an application for a patent (202110440791.X) on April 23, 2021.

### Data Availability
The raw data and statistical analysis and the raw data/uncropped Gels for Western blot analysis are available in the Supplemental Files.

### Supplemental Information
Supplemental information for this article can be found online at http://dx.doi.org/10.7717/peerj.13619#supplemental-information.

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
