# Peer review of "Agrochemical control of gene expression using evolved split RNA polymerase"

_PeerJ, doi:10.7717/peerj.13619_

## Round 0.1 · original submission · Major Revisions

Dear authors:

We have received the expert opinion of two reviewers that find your study interesting, but some concerns have been pointed out. Reviewers 1 and 2 have concerns regarding the experimental design and some controls are missing. The figure legends are not that informative and need clarification throughout the manuscript.

I kindly request that you carefully revise the content of the manuscript, address all the concerns point by point and submit a revised version of your manuscript.

Thank you so much for submitting your work to PeerJ.

Best regards
Bernardo

Reviewer 1 ·

Basic reporting

In this manuscript, the authors use the traditional split T7 RNA polymerase in combination with the ABA receptor PYRABACTIN RESISTANCE 1 (PYR1), a receptor that is known to interact with the chemical mandipropamid to reconstitute transcription. The manuscript is a standard manuscript in the field of split T7 RNA polymerases in which the authors have a ligand that binds to a protein and stabilizes T7 RNA polymerase and therefore transcription can be started.
The main concern that this reviewer has with this manuscript is the absence of crude data in the main figure, for instance, a sequencing denaturing acrylamide gel showing the accumulation of transcript of an SDS gel showing the accumulation of GFP (although this data is present in the supplementary figure).
In the opinion of this reviewer, the authors show that coupling of the agrochemical mandipropamid to a modified receptor can reconstitute T7RNAP and therefore the manuscript should be accepted.

Experimental design

The experimental design is standard for the field.

Validity of the findings

Data validation is standard for the field.

Reviewer 2 ·

Basic reporting

Yuan and Miao describe the use of mandipropamid inducible T7 RNA polymerase system in E. coli which could also be utilized together with plant systems. The study could be of interest to researchers, however, the overall description of experimental results are lacking details and clarity. The authors need to improve their manuscript through additional experiments and clarifications.

Experimental design

The most important finding of the manuscript is mandipropamid-inducible T7 RNAP system. The description is too brief and lacking details. They need to provide additional experiments including in vitro assay to test association of the split T7 RNAP system.

They also discuss split-GFP based system, but the description is short and lacks clarity.

For figures, they use connecting lines without describing what they are. The curve do not look like they were fit to data. They should remove those lines unless they mean something. For experiments, they do not report the number of replicates nor the statistical tests.

Validity of the findings

Overall, they need to provide detailed experimental setup with additional images. For instance, it is difficult to understand the setup for E. coli grown in hollow stems of water spinach. If the stems of water spinach were simply used as a container, how would it help them study plant-microbiome interaction as they proposed? They could use microscope images of sections to better understand the plant and microbe interaction under the culture condition.

Additional comments

There are very few references and they seem to miss a number of relevant literature on the topic.

---

## Round 0.2 · accepted · Accept

Dear authors,

After receiving your revised manuscript, I am very happy to announce that the two experts have found the required clarifications and improvements and now the manuscript is suitable for publication. Thank you for addressing the comments done by the reviewers and for choosing Peer J for submitting your work.

WIth best regards,
Bernardo

Reviewer 1 ·

Basic reporting

The authors have addressed all the concerns raised by this reviewer

Experimental design

No comment

Validity of the findings

No comment

Additional comments

No comment

Reviewer 2 ·

Basic reporting

The authors improved the manuscript with additional experiments and explanations in the main text. They also included a number of relevant literature to put their work in the context of existing literature. Thus, I would like to recommend the work for publication.

Experimental design

no comment

Validity of the findings

no comment

Additional comments

no comment